# Quality Improvement in Mackerel Fillets Caused by Brine Salting Combined with High-Pressure Processing

**DOI:** 10.3390/biology11091307

**Published:** 2022-09-02

**Authors:** Chih-Hsiung Huang, Chung-Saint Lin, Yi-Chen Lee, Jhih-Wei Ciou, Chia-Hung Kuo, Chun-Yung Huang, Chih-Hua Tseng, Yung-Hsiang Tsai

**Affiliations:** 1Department of Fisheries Production and Management, National Kaohsiung University of Science and Technology, Kaohsiung 811213, Taiwan; 2Department of Food Science, Yuanpei University of Medical Technology, Hsin-Chu 30015, Taiwan; 3Department of Seafood Science, National Kaohsiung University of Science and Technology, Kaohsiung 811213, Taiwan; 4Department of Fragrance and Cosmetic Science, Kaohsiung Medical University, Kaohsiung 807378, Taiwan

**Keywords:** hurdle technology, high hydrostatic pressure, mackerel fillet, brine salting, microbial inactivation

## Abstract

**Simple Summary:**

High-pressure processing enhanced the pasteurization effect of mackerel fillets that had undergone brine salting and improved the effect of texture hardening and fish discolouration observed after the high-pressure treatment alone. Our findings provide seafood processors as an alternative option to improve the safety of salt-brined mackerel fillets.

**Abstract:**

The purpose of the study is to investigate the effects of brine salting and high-pressure processing (HPP) on the microbial inactivation and quality parameters of mackerel fillets. Mackerel fillets were immersed in 3% and 9% sodium chloride brine for 90 min at refrigerator temperature, and then treated at 300, 400, 500, and 600 MPa pressure for 5 min. The microbial counts and physicochemical qualities of the fish were examined. In comparison with fish fillets treated with brine or high pressure alone, those treated with the combination of brine salting and HPP showed significantly reduced aerobic plate count (APC) and psychrotrophic bacteria count (PBC). The hardness and chewiness of salt-brined fillets were obviously lower than those of the unsalted fillets under the same pressure condition. Thus, brine salting imparted mackerel fillets a softer texture, which compensated for the HPP-induced increased hardness and chewiness of the fillets. The *L** (lightness) and Δ*E* (colour difference) values of the fillets increased with increasing pressure, with or without brine salting. Conversely, *a** (redness) values decreased with increasing pressure. The samples treated with 3% brine in combination with 300 or 400 MPa pressure had *a** values similar to those of the samples processed under similar HPP conditions alone but showed lower Δ*E* values than the other groups. Therefore, as a very high pressure would adversely affect the texture and colour of the fish fillets, this study suggests that immersion in an appropriate brine concentration (3%) and treatment with HPP at 400 MPa for 5 min improved or maintained the colour and texture relatively well and produced a synergistic bactericidal effect.

## 1. Introduction

High-pressure processing (HPP) is a novel non-thermal processing technology for food. It usually refers to the inactivation of spoilage bacteria and pathogenic bacteria in food with a pressure of at least 300 MPa or more to achieve the purpose of food safety and extending the storage period of food [1,2]. Because HPP-processed foods can retain more nutrients and aroma substances than heat-processed foods, this processing method is often used in the processing of heat-sensitive foods. Recently, HPP has usually been broadly applied in the production of fruit drinks and meat, vegetables, and seafood [3]. HPP can denature muscle proteins and connective tissues by destroying the tertiary protein structure without affecting the flavour of shellfish and release the adductor muscle to reach the shucking purpose. Therefore, HPP technology is also widely used in the shucking of shellfish [4]. Moreover, it can affect the gelling properties of seafood products. However, the application of HPP technology on fish meat can have several disadvantages, including change in colour (i.e., higher *L** and lower *a** values) accompanied by higher opacity and firmer texture of the final product [5]. In general, the *L** (lightness) parameter increases in pressurized fish, which is clearer, grey, typical of cooked meat aspect. The increase in *L** values depends only on the intensity of HPP treatment (pressure and time), and may be associated with globin and myofibrillar denaturation [2,4]. In addition, most studies have shown the decrease in *a** (loss of redness) in fish meat, which varies with the species and the pressurization parameters. The mechanism of protein (globin) denaturation with possible heme release can lead to *a** change [2,4]. With regard to texture, hardness obtained in the compression force test is one of the primary endpoints, with an increase observed in HPP-treated fish meats [5].

Salting is an ancient food processing method that uses a high sodium chloride (NaCl) concentration to generate osmotic pressure that ruptures bacterial cells, thereby inhibiting or killing food pathogens and spoilage bacteria. This processing method is often used to retain the quality and extend the shelf-life period of food. It is especially suitable for fish products that are perishable and whose preservation is difficult to maintain [6]. Additionally, salting plays a key role in improving the texture characteristics (i.e., tenderness and juiciness) of meat products [7]. However, as people’s eating habits change, lightly salt-brined products are gaining popularity among consumers. Hence, aquatic product processors often use brine salting to make lightly brined products [8]. Recently, hurdle technology that combines several physical and chemical treatments has turned into a new tendency in food processing and is often used to replace chemical preservatives and biocides [9]. If the hurdle heights (i.e., the strength of the preservation factor) in the food were moderately increased, the impact of microorganisms in food could be efficiently inactivated to fulfil the requirements of food hygiene and safety [9]. Commonly used hurdle technologies in food products include sugaring, salting, pH adjustment, low temperature storage, water activity reduction, high pressure treatment, and the addition of biocides/preservatives. In a recent study, the count of *Listeria monocytogenes* inoculated into minced chicken containing 1.5–2.5% calcium chloride (CaCl_2_) and processed under high pressure at 600 MPa for 60 s reduced by 1.12–1.21 logarithm as compared to that in minced chicken subjected to HPP treatment alone [10]. HPP and NaCl have been combined in meat processing in numerous studies. For instance, Ros-Polski et al. [7] proposed that a low-concentration NaCl salting combined with HPP improved the texture and colour of white chicken meat. Furthermore, it has been reported that HPP can be used to improve the microbial safety in reduced salt (NaCl)-treated minced chicken [10]. Crehan et al. [11] pointed out that processing with HPP (150 and 300 MPa) improved the functional properties of low-salt frankfurters.

Spotted mackerel (*Scomber australasicus*) is a broadly spread fish in the Indo-West Pacific Ocean. Chub mackerel and spotted mackerel are two principal mackerel species that are caught in the waters off Taiwan, spotted mackerel accounting for 78% of the total mackerel catch [12]. After being caught, it is usually canned, salted, or frozen [12]. The majority of the catch was mainly processed into salted fish for consumption [13]. Lately, the increase in the awareness of health requirements and changes in taste among consumers have made salted mackerel fillets prepared by the brine-salting method gradually popular [14]. We discovered in our previous research that the HPP treatment of spotted mackerel fillets at 500 and 600 MPa significantly reduced the aerobic plate count (APC) and psychrotrophic bacterial count (PBC) to undetectable levels. However, HPP might exert some adverse effects on the colour and texture of fish meat [13]. Recently, the results of our previous study showed that brine salting at a proper brine concentration (3%) and followed by high-pressure processing at 300 or 400 MPa for 5 min can improve or maintain a relatively good colour and texture, as well as result in a synergistic bactericidal effect [15]. However, to date, there has not been any study of the combination of high-pressure processing and brine salting on mackerel, which is commonly eaten by humans. Therefore, the novelty of this study is mainly to investigate how the brine-salting treatment on mackerel fillets prior to HPP improve the qualities caused by high pressure.

The principal aim of this study is to evaluate the effects of HPP, in combination with brine salting at high or low NaCl concentrations, on the microbial counts and physicochemical properties of mackerel fillets. In this study, mackerel fillets were immersed in 3% and 9% brine at 4 °C for 90 min and then processed under 300, 400, 500, and 600 MPa for 5 min to observe the impacts in microbial and physicochemical characteristics. In this experiment, a high salt concentration of 9% was used to simulate the conventional processing method of salted mackerel, while a low salt concentration of 3% was used to fulfil the current trend among people who prefer lightly salted food [8].

## 2. Materials and Methods

### 2.1. Preparation of Mackerel Fillets

All spotted mackerel (*Scomber australasicus*) caught in the waters off the Pacific coast of Taiwan, were transported in ice on fishing boats to Kaohsiung fishery port and market. In order to ensure the best uniformity of fish quality, twenty-three fresh spotted mackerel (*Scomber australasicus*), with a mean length of 35 ± 3 cm and weight of 380–400 g, were purchased from the same batch in this Kaohsiung fishery port and market, Taiwan, and instantly transported over trash ice within 1 h to the innovative food processing Lab. of National Kaohsiung University of Science and Technology. Here, the fish were sliced and skinned, and two pieces of fish fillets from dorsal region were obtained. Each fish fillet was 17 ± 3 cm long, 6 ± 2 cm wide, and 2 ± 0.5 cm thick and weighed 125–130 g. A total of 45 fish fillets were divided into three groups, including the control group (group not subjected to high pressure), the HPP treatment group, and the combination of brine salting and HPP treatment group. The control group comprised raw fish fillets (unbrined), 3% NaCl brined samples, and 9% NaCl brined samples. The HPP treatment group comprised samples of fresh fish fillets pressurised at 300, 400, 500, and 600 MPa for 5 min. For the brine salting combined with HPP treatment group, fresh fish fillets were immersed in 3% and 9% brine and then pressurised for 5 min under 300, 400, 500, and 600 MPa. The experiment was conducted in triplicates for each experimental group. In the brine-salting treatment process, fish fillets were immersed in 3% and 9% brine at a fillets to brine ratio of 1:2 (*w/v*) at 4 °C for 90 min. Then, brined fish fillets were placed on a clean stainless steel wire mesh to let the brine to drip dry for next experiments.

### 2.2. High-Pressure Processing Treatment

The each raw (non-salted) or salted piece of fillet was individually packaged into vacuum bags (NY/LLDPE), vacuumed and heat sealed for HPP treatment. The high-pressure processing facility (KeFa High-Pressure Technology, Baotou, Inner Mongolia, China) has a volume of 6.2 L, maximum operating pressure of 600 MPa, pressure ramp rate of approximately 45 MPa/s, and depressurization time of <10 s. The deionised H_2_O of 22 °C was used as the pressure-transmitting medium. The fish fillets were set to pressurize at 300, 400, 500, and 600 MPa for 5 min, but does not include the time of pressure increase and release. Lin et al. [13] found that, after the HPP treatments at 300 to 500 MPa for 5 min, the APC of mackerel fillets could be effectively reduced, so these parameters of pressures and time in this study were used; therefore, the high pressure conditions were set at 200, 300, 400, 500, and 600 MPa for 5 min.

### 2.3. Analysis of Microbial Counts

For the aerobic plate count (APC) analysis, fillet samples (10 g) were mixed with sterile saline (90 mL), followed by homogenization at 5000 rpm for 100 s using a sterilized blender (Omni International, Waterbury, CT, USA) and then 1.0 mL homogenous solution was added to 9.0 mL sterile saline for serial ten-fold dilutions. The homogenate and serial ten-fold diluted solution (0.1 mL) were spread on Tryptic soy agar (TSA) in duplicate. Thereafter, the plates were placed in incubator at 35 °C for 24–48 h, and the colony numbers on the plate were counted. The counts were expressed as average ± standard deviations of the data obtained from three independent fillet samples [16]. For psychrotrophic bacteria count (PBC) analysis, the homogenate and serial ten-fold diluted solution (0.1 mL) were spread on TSA in duplicate and then incubated at 7 °C for 10 days. Finally, the colony numbers on the TSA dish were calculated [16].

### 2.4. Texture

A TA.XT2 texture analyser (Stable Micro System Ltd., Surrey, UK) was used to perform the texture profile analysis (TPA) of fillet samples after treatment, which included determinations of hardness, cohesiveness, springiness, and chewiness [13]. The TPA was performed by two-cycle compression tests using a round TA18 probe (12.7 mm in diameter) with a target distance of 5.0 mm, a predicted speed of 2 mm/s, a test and return speed of 1.5 mm/s, and a trigger point load of 0.1 g. The compression rate was 50% of the original height, the pressing depth was 1.0 cm, and the holding time was 3 s. Measurements were obtained from three points on each fish fillet, which were placed on a support base with the probe inserted perpendicularly. Three individual fillets were carried out from each treatment (triplicates). The obtained average values of each treatment were used as the final result.

### 2.5. Colour

To ascertain the colour of fish fillets subjected to the different treatments, a CR-300 Chroma meter colorimeter (Konica Minolta, Inc., Tokyo, Japan) was used to perform colour analysis [13]. Prior to colour measurements, the colorimeter was calibrated using standard white (*L** = 96.72, *a** = 0.11, *b** = −0.14) and red (*L** = 51.13, *a** = 50.00, *b** = 24.03) plates. Following calibration, three colour measurements at different locations on the fish fillets were performed, at each of which, the lightness (*L**), redness (*a**), and yellowness (*b**) values were recorded. On the basis of these values, values for colour difference (ΔE) were calculated using Equation (1) and the results are presented as average values [17].
Δ*E* = [(*L*_2_** − L*_1_***)^2^ + (*a*_2_** − a*_1_***)^2^ + (*b*_2_** − b*_1_***)^2^]^1/2^(1)
where *L_1_**, *a_1_**, and *b_1_** are original or brine-salted fillet colorimetric values and *L_2_**, *a_2_**, and *b_2_** are the colorimetric values of the fillet samples after HPP or brine salting combination.

### 2.6. Total Volatile Basic Nitrogen (TVBN)

Determinations of the total volatile basic nitrogen (TVBN) content of fish samples were based on the Conway microdiffusion dish procedure proposed by Cobb et al. [18]. Initially, a 5 g sample of fish was placed in a 50 mL centrifuge tube, to which was added 20 mL of 6% trichloroacetic acid (TCA). The tube was placed in an Ultra-Turrax IKA T18 basic homogenizer (IKA-Werke GmbH & Co. KG, Staufen, Germany) and homogenized at 5000 rpm for 1.5 min. The resulting homogenate was centrifuged at 3000× *g* for 8 min at 4 °C and the supernatant thus obtained was filtered through Whatman Grade 1 filter paper. This procedure was repeated twice and the final filtrate was collected and made up to 50 mL with 6% TCA. A 1 mL aliquot of this filtrate and 1.0 mL of saturated K_2_CO_3_ solution were placed in the outer ring of a Conway dish, followed by the addition of 1.0 mL of boric acid solution to the inner ring. Having covered the dish, it was placed in a 45 °C incubator for 60 min, after which, the boric acid solution in the inner ring was end-point titrated with 0.02 N HCl to determine TVBN values (expressed in terms of mg/100 g of fish).

### 2.7. Salt and Moisture Contents

A finely chopped fish fillet sample (5.0 g) was put in a crucible, heated in an ashing furnace at 550 °C for 12–14 h to give the sample a grey or off-white colour, and then set in a desiccator to let it to cool to room temperature. The ashing sample was diluted to 100 mL with deionised H_2_O and mixed well. The salt content was determined using precipitation titration (Mohr’s method) [19]. 

The moisture content was examined by the standard gravimetric method [19]. A 1–3 g sample of finely minced fish fillet was weighed after drying in an oven at 105 °C for 4 h. The moisture content was calculated after further drying for 1.5 h or longer until the sample weight did not change by more than 0.5 mg [19].

### 2.8. Statistical Analysis

For data analysis, the SPSS software (SPSS for windows, version 21, IBM Corp., Armonk, NY, USA) was used to perform it. The data obtained were subjected to a one-way analysis of variance (ANOVA), with Tukey’s test being used to determine differences in the mean values among triplicate samples. A *p*-value of less than 0.05 was considered indicative of a statistically significant difference.

## 3. Results and Discussion

### 3.1. The Microbial Counts of Mackerel Fillet

Table 1 shows the results of APC in mackerel fillets subjected to brine salting combined with HPP treatment. The initial APC of fresh fish fillets was 4.82 log CFU/g, which was reduced to 4.63 and 4.05 log CFU/g (i.e., a reduction of 0.19 and 0.77 log CFU/g, respectively) after being brined at 3% and 9% salt concentrations, respectively. This result indicates that the brine-salting treatment alone slightly reduced the APC count of the fish fillets and that the effect was better at a higher brine concentration (9%). This is probably because brine helps to wash away the bacteria adhering on the surface of fillets or inhibits their growth in fillets, especially at high salt concentrations. A similar trend was showed in the HPP treatment group. The APC of the samples processed with 3% and 9% brine combined with 300 MPa HPP reduced from 4.72 log CFU/g at 300 MPa high pressure alone to 4.23 and <2.0 log CFU/g, respectively (i.e., a reduction of 0.59 and 4.82 log CFU/g, respectively). The APC of the samples treated with the 3% and 9% brine combined with 400 MPa HPP reduced from 2.32 log CFU/g at 400 MPa high pressure alone to undetectable levels (<2.0 log CFU/g) (i.e., a reduction of 4.82 log CFU/g). In particular, the APC was zero when the samples were pressurized at 500 and 600 MPa alone or in combination with the 3% and 9% brine treatments. Moreover, the APC of fillet samples significantly decreased as the pressure increased with or without brine salting treatment. Lin et al. [13] reported similar results where the higher the pressure (300–600 MPa, 5 min) on the processed mackerel fillets, the lower APC was observed. The APC was zero in the mackerel fillets subjected to a pressure treatment of 500 and 600 MPa. In summary, the combination of brine salting and high pressure (300 and 400 MPa) to treat mackerel fillets exhibits an additive antimicrobial effect and that HPP in combination with brine salting might be considered as an efficient hurdle technology.

Table 2 shows the results of PBC in mackerel fillets treated with brine salting and high pressure. As observed with APC, the PBC of raw mackerel fillets was 4.66 log CFU/g but decreased to 4.45 and 4.10 log CFU/g (i.e., a reduction of 0.21 and 0.56 log CFU/g, respectively) after the 3% and 9% brine-salting treatment, respectively. This result indicates that the brine-salting treatment alone reduced the PBC in fish fillets and that the effect was more pronounced at a higher brine concentration (9%). This may be because brine helps to wash off the psychrotrophic bacteria adhering on the surface of fillets or inhibits their growth in fillets, especially at high salt concentrations. The PBC of the samples treated with the 3% and 9% brine combined with 300 MPa HPP treatment reduced from 4.54 log CFU/g at 300 MPa HPP only to 4.29 and <2.0 log CFU/g, respectively (i.e., a reduction of 0.37 and 4.66 log CFU/g, respectively). The PBC of the samples processed with the 3% and 9% brine in combination with HPP at 400 MPa reduced from 2.11 log CFU/g at 400 MPa HPP alone to undetectable levels (<2.0 log CFU/g) (i.e., a reduction of 4.66 log CFU/g). In particular, the PBC was zero when the samples were autoclaved at 500 and 600 MPa alone or in combination with the 3% and 9% brine-salting treatment. Therefore, the results of this study show that, under similar pressure conditions, a higher brine concentration reduced the PBCs in fish fillets. Furthermore, under similar brine-salting conditions, the PBCs in fish fillets significantly reduced as the pressure increased. Overall, brine salting in combination with high pressure (300 and 400 MPa) treatment was more effective in eliminating PBCs in mackerel fillets than either the single treatment with brine salting or HPP. This may be because the osmotic pressure generated by the sodium chloride solution causes the instability of the bacterial cell membrane and wall, and the high pressure causes the ruptures and holes in the cell membrane or wall. The additive impact of both treatments is the reason of the significant reduction in the number of microorganisms.

These results are also in line with those reported by Tsai et al. [15] who treated milkfish meat with the 3% and 9% NaCl brine combined with 300–600 MPa pressure for 5 min; the APC and PBC in fish meat reduced as brine concentration and pressure level increased, suggesting that the combination of brine salting and HPP treatment had a synergistic bactericidal effect. The mechanism underlying the NaCl-mediated inhibition of the growth of microorganisms is mainly associated with the reduction in the water activity in food. This phenomenon results in the inhibition of the nutrient delivery to cells, osmotic pressure imbalance, and changes in the swelling pressure [10]. HPP can cause the irreversible denaturation of enzymes and proteins in bacteria and rupture the cell membrane or cell wall, resulting in the expulsion of internal substances and ultimately bacterial death [1]. It has been reported that HPP can reduce protein content in cells by destroying the protein structure, reducing protein biosynthesis, and inhibiting protein repair, eventually leading to bacterial damage or death [2].

### 3.2. The Texture of the Mackerel Fillet

Table 3 shows the results of texture changes in mackerel fillets after the combination treatment of brine salting and HPP. The hardness of the samples immersed in the 3% and 9% brine was significantly lower than that of fresh fish fillets (*p* < 0.05). The hardness after pressurisation significantly increased as the pressure increased with or without brine salting. The hardness of fresh samples and the 3% and 9% brine salted samples increased from 1.16, 0.76, and 0.71 (N) to 5.52, 5.61, and 5.86 (N) in the 600 MPa group, respectively. The hardness of the group treated with the 3% and 9% brine combined with the 300–500 MPa HPP treatment was significantly lower than that of the group treated under similar pressure alone. However, there was no significant difference in cohesiveness between brined and unbrined fish fillets. In general, the cohesiveness of the fish fillets slightly increased with an increase in pressure, regardless of whether it was brined or not. Food cohesiveness indicates the degree to which the food deforms by force before it breaks or crumbles. Hence, high cohesiveness points out that the constituents of the fish meat can better adhere together. The springiness of the HPP-treated samples significantly increased and was the highest at 300 MPa and subsequently decreased with any further increase in pressure. Similar results were obtained with hardness in terms of chewiness, as both the 3% and 9% brine salted samples had significantly lower chewiness than that of the fresh fish fillets (*p* < 0.05). The chewiness of the pressurised samples significantly increased with the increase in pressure, irrespective of brine-salting treatment. The chewiness of fresh and the 3% and 9% brine-salted samples increased from 7.45, 7.23, and 7.33 (mJ) to 28.84, 26.49, and 26.67 (mJ) at 600 MPa, respectively. The chewiness of the 3% and 9% brine salted samples treated with 300–600 MPa HPP was obviously lower than that of the fillets subjected to only HPP.

In general, the hardness and chewiness of the unbrined fillets were obviously higher than those of the brined fillets under similar pressure. On the other hand, no significant difference was observed in cohesiveness between fillets regardless of the brine-salting treatment under similar pressure conditions. This result exhibits that brine salting compensated for the increased hardness and chewiness of mackerel fillets resulting from high pressure and led to a softer texture of samples. Consequently, the texture of the mackerel fillets can be improved or retained relatively well when treated with appropriate brine concentration in combination with HPP. The results of this study were similar to our previous report in which milkfish meat treated with the 3% and 9% brine combined with high pressure had reduced hardness and chewiness as compared with the meat treated with HPP alone [15].

Ros-Polski et al. [7] observed similar results and found that a low-concentration NaCl brine salting combined with a high pressure treatment could improve the texture of white chicken flesh. Pszczola [20] also proposed that NaCl activated the muscle protein that increased protein hydration and water-binding capacity, thereby improving the tenderness and juiciness of processed meat products. In addition, the increase in hardness of fish meat samples after HPP might be related to the denaturation and aggregation of myofibrillar proteins, which could lead to the shrinkage of tissue structures under high pressure [21]. Researchers have reported that the hardness of cod fish meat increases with an increase in pressure after 2 min of processing at 200–500 MPa [22]. Similar results were presented in our previous study [13] in which the hardness and chewiness of the mackerel fillets increased with an increase in pressure. However, after the HPP treatment, the reduction in the textural property or the possible softening effect related to protein aggregation and the fragment of myofibrils structures can also be observed in seafood [22]. This study indicated that 90 min of brine salting might lead to the protein solubilisation of fish and help to reduce the hardening effect of high pressure on mackerel fillets caused by protein aggregation. Overall, this study found that the 3% brine salting combined with pressures of 300 and 400 MPa groups were the better treatment condition, which had lower hardness and chewiness values compared to other groups.

### 3.3. The Color Values of Mackerel Fillet

Table 4 shows the resultant colour changes in mackerel fillets after treatment with brine and HPP. The *L** value (lightness) of fish fillets significantly increased after the 3.0% and 9.0% brine-salting treatment from 57.06 (raw fish fillets) to 61.01 and 60.39, respectively. There was a significant increase in the *L** value of fillets as the pressure increased, regardless of brine-salting treatment. Under similar pressure conditions, no significant difference was observed between the *L** values of the fish fillets treated with or without brine salting. On the contrary, the *a** values (redness) decreased as the pressure increased. There was no significant difference between the *a** values of the fish fillets treated with or without brine salting in the control, 300 MPa, and 400 MPa groups. On the contrary, the *a** values of the fillets by HPP alone from the 500 and 600 MPa groups were significantly higher than those of the 3% and 9% brine-salted fillets (*p* < 0.05). A similar trend was observed in *b** values (yellowness) regardless of brine salting, i.e., the *b** values of the fillets after HPP treatment decreased with an increase in pressure. In addition, the Δ*E* values increased with an increase in pressure. The Δ*E* values of the fillets treated with 3% brine combined with 300 and 400 MPa pressure treatment were 1.83 and 2.28, respectively, which were significantly lower than the values reported for other HPP treatment groups (Table 4). These results are consistent with those reported in previous studies demonstrating that milkfish meat treated with 3% brine in combination with 300 or 400 MPa high pressure maintained higher *a** and lower Δ*E* values as combined with similar pressure alone or with 9% brine as compared to similar HPP treatments [15].

The brined fillets from the control group (with higher *L** values) were brighter than unbrined fillets. A study also indicated that salting can lead to an increase in the *L** value of grass carp meat and that the brightness of grass carp meat can be enhanced when salted at a low-salt concentration for a short duration (lightness) [23]. It has been suggested that the calcium and magnesium ions in salt can turn the appearance of salted cod muscle white [24]. A previous study found that white chicken meat treated with 1.5–2.5% NaCl had higher *a** values than unbrined white chicken meat [7]. The addition of low concentrations of NaCl to turkey meat increased its *a** value, possibly owing to the solubilisation of myofibrillar proteins that reacted with heme pigments [25]. Nevertheless, in this study, the high concentration brine (9%) group had the lowest *a** value and showed significant discoloration (reduced redness) as compared to the unbrined and 3% brine treatment groups. A plausible reason for this observation could be that a high concentration brine salting might cause the dehydration of fish meat, leading to the denaturation and loss of water-soluble proteins containing heme pigments, eventually causing the discolouration of fish meat [23]. Moreover, the changes in the colour parameters of fish meat after HPP treatment were attributed to the denaturation of myofibrillar and sarcoplasmic proteins [26]. The degradation of the pigment and myoglobin of fish meat caused by the high-pressure treatment was the main reason for the reduction in the redness of fish meat [27].

Overall, the samples treated with 3% brine in combination with 300 and 400 MPa pressure had lower Δ*E* values and similar *a** values than those treated with 9% brine combined with HPP or HPP alone. Therefore, treating mackerel fillets with a low concentration (3%) of brine combined with appropriate pressure (300 and 400 MPa) can preserve their colour.

### 3.4. The Total Volatile Basic Nitrogen (TVBN), Salt, and Moisture Contents of Mackerel Fillets

Table 5 shows the results of the changes in TVBN values of the mackerel fillets treated with the combination of brine salting and HPP. The initial TVBN of the fresh mackerel fillets was 5.27 mg/100 g, which changed to 5.58 and 4.90 mg/100 g after salting in 3% and 9% brine, respectively (minimal change). Similarly, the TVBN values of the 300–600 MPa high-pressure-treated samples were between 4.85 and 5.65 mg/100 g, regardless of brine salting, and were not statistically different from each other (*p* > 0.05).

Table 6 shows the results of the changes in the salt content and moisture content of the mackerel fillets processed by brine salting combined with high pressure. The salt content of raw fish fillets and those treated with high pressure alone (300–600 MPa) ranged from 0.35% to 0.39% and showed no significant difference. This result indicates that HPP only has no effect on the salt content of the fresh fillet. Additionally, the salt content of the fillets increased to 0.93% and 1.88% after the 3% and 9% brine-salting treatment, respectively. However, the salt content of the samples tended to slightly decrease after the high-pressure (300–600 MPa) treatment. The salt content of the group treated with 3% brine salting and HPP group ranged between 0.79% and 0.86%, and no significant difference was observed between them. Likewise, the salt content of the group treated with the combination of 9% brine and HPP ranged from 1.77% to 1.82% and did not significantly differ between the groups.

The moisture contents of the fresh mackerel fillets and the samples salted in 3% and 9% brine were 49.33%, 57.43%, and 56.09%, respectively, which indicated that the fish absorbs water and swells after being immersed in brine. The moisture content of the fish fillets treated with high pressure (300–600 MPa) alone ranged from 48.92% to 50.61% and did not significantly differ between the groups. However, the moisture content of the samples treated with 3% and 9% brine in combination with HPP (>400 MPa) showed a decreasing trend with an increase in the pressure and reached a minimum value of 46.92% and 47.82% at 600 MPa, respectively. Overall, high pressure in combination with brine salting reduced the salt content and moisture content of mackerel fillets that further reduced at a higher pressure. This observation might be related to the breaking of non-covalent bonds between molecules (such as hydrogen bonds and hydrophobic bonds) at a high pressure, which eventually denatures the proteins in the fish muscle and reduces the water-binding capacity, resulting in the loss of drips, moisture content, and salt content. A similar study was also conducted by Tsai et al. [15] who indicated that HPP treatment reduced the moisture and salt content of brined milkfish meats where the reduction was more pronounced as the pressure increased.

The majority of previous studies on high pressure in combination with NaCl treatment have focused on using the high-pressure treatment on meat and meat products as a substitute processing to lower the level of NaCl used; this process not only reduced the amount of microorganisms but also maintained the physical characteristics and quality of meat products [7,28,29]. To our knowledge, this is the first study that analysed the effect of the combined treatment of brine salting and HPP and highlighted the improvement in the physicochemical characteristics and microbiological safety of mackerel fillets. These findings suggest that brine salting compensated for the HPP-induced increase in the hardness and chewiness of mackerel fillets and imparted a softer texture. In addition, 3% brine salting in combination with 300 or 400 MPa HPP treatment lower the Δ*E* value than other treatments and maintained the *a** value similar to that observed after the pressure treatment alone. Furthermore, the brine salting alone reduced the APC and PBC of fish fillets, and this effect was more pronounced at a higher pressure or brine concentration. In particular, the microbial count was zero in the fish fillets treated with the combination of 3% brine and 400 MPa high pressure. Considering all these factors, the optimum processing conditions were mackerel fillets salted in 3% brine and 400 MPa pressure for 5 min.

Currently, the use of HPP by food industries and the sale of pressurized products is a reality, ranging from fruits and vegetables to seafood and eggs, with wide consumer acceptance [30]. In addition, the Food and Drug Administration (FDA) and US Department of Agriculture (USDA) have approved the technology as a food preservation method, and the US National Advisory Committee on Microbiological Criteria for Foods regards HPP as a non-thermal pasteurization process that can replace conventional pasteurization [2]. Even though HPP requires high-pressure equipment, which is expensive and difficult to manufacture, mainly in relation to the high initial capital to be invested, it is a clean technology since it presents a significantly lower carbon footprint than thermal methods [2,30].

## 4. Conclusions

The results of this study exhibit the synergistic bactericidal effect of the brine salting combined with high pressure treatment on APCs and PBCs. Brine salting compensated for the increased hardness and chewiness of the fillets resulting from high pressure processing and imparted a softer texture to the product. The *L** and Δ*E* values of mackerel fillets increased with increasing pressure regardless of brine salting, whereas the *a** values demonstrated the opposite trend. Therefore, the appropriate selection of processing methods combining high pressure and brine salting is able to be considered a successful hurdle method for improving the quality of mackerel fillets. Further research is required to study the storage life of mackerel fillets after treating the combination methods. Information generated from this study also provide insights into HPP impacts on salted mackerel fillets, and these data are very important for the development of further industrial applications of this novel, non-thermal, fresh seafood-processing technology to replace conventional methods.

## Figures and Tables

**Table 1 biology-11-01307-t001:** The aerobic plate count (log CFU/g) of mackerel fillets after the brine salting of 3.0% and 9.0% for 90 min at 4 °C and then pressurization at 300, 400, 500, and 600 MPa for 5 min.

Brine Concentration (%)	Control (0.1 MPa)	HPP Treatment (MPa)			
		300	400	500	600
Raw fillet	4.82 ± 0.08 *^1aA^	4.72 ± 0.06 ^aA^ (0.10) *^2^	2.32 ± 0.10 ^b^ (2.50)	<2.0 (4.82)	<2.0 (4.82)
3.0	4.63 ± 0.06 ^aB^ (0.19)	4.23 ± 0.08 ^bB^ (0.59)	<2.0 (4.82)	<2.0 (4.82)	<2.0 (4.82)
9.0	4.05 ± 0.15 ^C^ (0.77)	<2.0 (4.82)	<2.0 (4.82)	<2.0 (4.82)	<2.0 (4.82)

*^1^ Data are average ± standard deviation for triplicate. Averages in every row with the different lowercase letter are significantly different (*p* < 0.05). Averages in every column with the different uppercase letter are significantly different (*p* < 0.05). *^2^: Reductions in bacterial populations (log CFU/g) after treatments as compared with raw fillet.

**Table 2 biology-11-01307-t002:** The psychrotrophic bacteria count of mackerel fillets after the brine salting of 3.0% and 9.0% for 90 min at 4 °C, and then pressurization at 300, 400, 500, and 600 MPa for 5 min.

Brine Concentration (%)	Control (0.1 MPa)	HPP Treatment (MPa)			
		300	400	500	600
Raw fillet	4.66 ± 0.05 *^1aA^	4.54 ± 0.07 ^b^^A^ (0.12) *^2^	2.11 ± 0.10 ^c^ (2.55)	<2.0 (4.66)	<2.0 (4.66)
3.0	4.45 ± 0.19 ^aB^ (0.21)	4.29 ± 0.14 ^bB^ (0.37)	<2.0 (4.66)	<2.0 (4.66)	<2.0 (4.66)
9.0	4.10 ± 0.19 ^C^ (0.56)	<2.0 (4.66)	<2.0 (4.66)	<2.0 (4.66)	<2.0 (4.66)

*^1^: Data are average ± standard deviation for triplicate. Averages in every row with the different lowercase letter are significantly different (*p* < 0.05). Averages in every column with the different uppercase letter are significantly different (*p* < 0.05). *^2^: Reductions in bacterial populations (log CFU/g) after treatments as compared with raw fillet.

**Table 3 biology-11-01307-t003:** The texture properties of mackerel fillets after the brine salting of 3.0% and 9.0% for 90 min at 4 °C, and then pressurization at 300, 400, 500, and 600 MPa for 5 min.

Texture Properties	Brine Concentration (%)	Control (0.1 MPa)	HPP Treatment (MPa)			
			300	400	500	600
Hardness (N)	Raw fillet	1.16 ± 0.39 *^1dA^	3.65 ± 0.16 ^cA^	3.85 ± 0.21 ^cA^	5.09 ± 0.08 ^bA^	5.52 ± 0.23 ^aA^
	3.0	0.76 ± 0.07 ^dB^	3.36 ± 0.11 ^cB^	3.60 ± 0.09 ^cB^	4.15 ± 0.33 ^bB^	5.61 ± 0.21 ^aA^
	9.0	0.71 ± 0.03 ^dB^	3.35 ± 0.16 ^cB^	3.50 ± 0.24 ^cB^	4.25 ± 0.20 ^bB^	5.86 ± 0.16 ^aA^
Cohesiveness	Raw fillet	0.69 ± 0.03 ^bA^	0.70 ± 0.03 ^bA^	0.77 ± 0.05 ^aA^	0.80 ± 0.05 ^aA^	0.75 ± 0.03 ^aB^
	3.0	0.66 ± 0.03 ^bA^	0.68 ± 0.07 ^bA^	0.73 ± 0.03 ^abA^	0.76 ± 0.03 ^abA^	0.81 ± 0.02 ^aA^
	9.0	0.64 ± 0.03 ^cA^	0.65 ± 0.04 ^cA^	0.71 ± 0.03 ^bA^	0.74 ± 0.07 ^bA^	0.86 ± 0.04 ^aA^
Springiness (mm)	Raw fillet	7.00 ± 0.15 ^bA^	9.90 ± 0.14 ^aA^	6.54 ± 0.06 ^cC^	6.55 ± 0.07 ^cB^	7.39 ± 0.22 ^bB^
	3.0	6.68 ± 0.22 ^dB^	10.13 ± 0.10 ^aA^	10.02 ± 0.14 ^aA^	9.07 ± 0.24 ^bA^	8.37 ± 0.13 ^cA^
	9.0	6.84 ± 0.30 ^cAB^	9.66 ± 0.27 ^aB^	9.22 ± 0.24 ^aB^	8.78 ± 0.15 ^bA^	8.16 ± 0.13 ^bA^
Chewiness (mJ)	Raw fillet	7.45 ± 0.07 ^cA^	24.52 ± 0.04 ^bA^	25.12 ± 1.65 ^bA^	25.62 ± 0.51 ^bA^	28.84 ± 0.97 ^aA^
	3.0	7.23 ± 0.05 ^dB^	22.48 ± 0.14 ^cB^	22.67 ± 0.29 ^cB^	24.21 ± 0.19 ^bB^	26.49 ± 0.26 ^aB^
	9.0	7.33 ± 0.05 ^cB^	23.31 ± 0.22 ^bB^	23.49 ± 0.37 ^bB^	24.12 ± 0.73 ^bB^	26.67 ± 0.11 ^aB^

*^1^: Data are average ± standard deviation for triplicate. Averages in every row with the different lowercase letter are significantly different (*p* < 0.05). Averages in every column with the different uppercase letter are significantly different (*p* < 0.05).

**Table 4 biology-11-01307-t004:** The *L**, *a**, *b**, and Δ*E* values of mackerel fillets after the brine salting of 3.0% and 9.0% for 90 min at 4 °C, and then pressurization at 300, 400, 500, and 600 MPa for 5 min.

Colour	Brine Concentration (%)	Control (0.1 MPa)	HPP Treatment (MPa)			
			300	400	500	600
*L**	Raw fillet	57.06 ± 0.48 *^1dB^	62.03 ± 0.17 ^cA^	62.09 ± 0.13 ^cA^	65.45 ± 0.26 ^bA^	66.10 ± 0.28 ^aA^
	3.0	61.01 ± 0.75 ^cA^	61.68 ± 0.42 ^cA^	62.25 ± 0.31 ^bA^	64.85 ± 0.52 ^aA^	65.82 ± 0.30 ^aA^
	9.0	60.39 ± 0.54 ^cA^	62.72 ± 0.35 ^bA^	63.19 ± 0.61 ^abA^	64.07 ± 0.24 ^aA^	65.78 ± 0.16 ^aA^
*a**	Raw fillet	6.43 ± 0.29 ^aA^	5.53 ± 0.36 ^bA^	4.96 ± 0.17 ^cA^	4.25 ± 0.03 ^dA^	4.11 ± 1.00 ^dA^
	3.0	6.48 ± 0.26 ^aA^	5.38 ± 0.18 ^bA^	4.90 ± 0.25 ^cA^	3.68 ± 0.35 ^dB^	3.59 ± 0.29 ^dB^
	9.0	6.07 ± 0.07 ^aA^	5.15 ± 0.19 ^bA^	4.47 ± 0.32 ^cA^	3.54 ± 0.34 ^dB^	3.51 ± 0.29 ^dB^
*b**	Raw fillet	15.25 ± 0.38 ^aA^	15.54 ± 0.45 ^aA^	15.15 ± 0.14 ^aA^	14.53 ± 0.16 ^bA^	14.42 ± 0.45 ^bA^
	3.0	16.09 ± 0.42 ^aA^	14.79 ± 0.39 ^bB^	14.99 ± 0.18 ^bA^	14.85 ± 0.32 ^bA^	15.07 ± 0.30 ^bA^
	9.0	15.54 ± 0.33 ^aA^	14.54 ± 0.38 ^bB^	14.93 ± 0.20 ^aA^	15.01 ± 0.25 ^aA^	14.70 ± 0.31 ^bA^
Δ*E*	Raw fillet	-	6.68	8.00	8.47	9.15
	3.0	-	1.83	2.28	4.97	6.48
	9.0	-	2.56	3.06	4.30	5.79

*^1^: Data are average ± standard deviation for triplicate. Averages in every row with the different lowercase letter are significantly different (*p* < 0.05). Averages in every column with the different uppercase letter are significantly different (*p* < 0.05).

**Table 5 biology-11-01307-t005:** The total volatile basic nitrogen (mg/100 g) of mackerel fillets after the brine salting of 3.0% and 9.0% for 90 min at 4 °C and then pressurization at 300, 400, 500, and 600 MPa for 5 min.

Brine Concentration (%)	Control (0.1 MPa)	HPP Treatment (MPa)			
		300	400	500	600
Raw fillet	5.27 ± 0.51 *^1aA^	5.53 ± 0.40 ^aA^	4.85 ± 0.26 ^aA^	5.55 ± 0.49 ^aA^	5.37 ± 0.62 ^aA^
3.0	5.58 ± 0.65 ^aA^	5.04 ± 0.39 ^aA^	4.90 ± 0.41 ^aA^	5.09 ± 0.39 ^aA^	4.99 ± 0.50 ^aA^
9.0	4.90 ± 0.46 ^aA^	4.99 ± 0.44 ^aA^	5.11 ± 0.15 ^aA^	5.65 ± 0.76 ^aA^	5.39 ± 0.38 ^aA^

*^1^: Data are average ± standard deviation for triplicate. Averages in every row with the different lowercase letter are significantly different (*p* < 0.05). Averages in every column with the different uppercase letter are significantly different (*p* < 0.05).

**Table 6 biology-11-01307-t006:** The salt and moisture contents (%) of mackerel fillets after the brine salting of 3.0% and 9.0% for 90 min at 4 °C, and then pressurization at 300, 400, 500, and 600 MPa for 5 min.

	Brine Concentration (%)	Control (0.1 MPa)	HPP Treatment (MPa)			
			300	400	500	600
Salt content	Raw fillet	0.38 ± 0.04 *^1a^	0.36 ± 0.02 ^a^	0.35 ± 0.05 ^a^	0.39 ± 0.02 ^a^	0.37 ± 0.03 ^a^
	3.0	0.93 ± 0.02 ^a^	0.85 ± 0.05 ^b^	0.86 ± 0.03 ^b^	0.81 ± 0.03 ^b^	0.79 ± 0.02 ^b^
	9.0	1.88 ± 0.01 ^a^	1.79 ± 0.03 ^b^	1.82 ± 0.03 ^b^	1.77 ± 0.02 ^b^	1.82 ± 0.03 ^b^
Moisture content	Raw fillet	49.33 ± 0.48 ^a^	50.20 ± 0.64 ^a^	48.92 ± 0.78 ^a^	50.61 ± 0.92 ^a^	50.49 ± 0.74 ^a^
	3.0	57.43 ± 0.78 ^a^	58.85 ± 0.96 ^a^	54.90 ± 0.81 ^b^	48.44 ± 1.61 ^c^	46.92 ± 0.86 ^c^
	9.0	56.09 ± 0.46 ^a^	57.25 ± 1.00 ^a^	55.12 ± 0.51 ^b^	55.07 ± 1.19 ^b^	47.82 ± 1.00 ^c^

*^1^: Data are average ± standard deviation for triplicate. Averages in every row with the different lowercase letter are significantly different (*p* < 0.05).

## Data Availability

Not applicable.

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
