# Peer review of "Quality Improvement in Mackerel Fillets Caused by Brine Salting Combined with High-Pressure Processing"

_biology, 2022, doi:10.3390/biology11091307_

Round 1
Reviewer 1 Report
The article “Hurdle effect of brine salting and high-pressure processing on microbial inactivation and quality improvement in mackerel fillets” is well written and can be accepted after minor revision.
Title: Good to have a shorter title.
Line 22 and 96: “3% or 9%”, if you have used both the concentrations, write “and”. Follow this everywhere.
Introduction: please provide information on similar studies carried out and the novelty of the research just before the aim of the article.
Materials and Methods:
It is not clear whether the fish samples were taken afresh? If not, mention about whether those were from same lot, what was storage condition and major changes in quality during primary storage.
Line 138: “Thereafter, after” remove “after”
Table 1: Title of the table is too much lengthy.
Results of the research are sufficiently justified giving appropriate reason for the trends. Authors should include one section on energy requirement and techno-economic feasibility of using HPP at commercial level.
In the conclusion, authors should include application of the research and future need of research.
Reviewer 2 Report
Generally, this manuscript investigated the hurdle effect of brine salting and high pressure processing on microbial inactivation and quality improvement in mackerel fillets. Mostly, the innovation of this work is doubted, since several researchers have reported their work about HPP and NaCl on meat matrix. And the attributes might be not enough to support the results.
And, some expression was not appropriate. For instance, in the abstract, L29-33 were talked about the color change during the treatments, thus I assume the color parameters were important in this study. However, there is only one sentence, L52-55, about color in the introduction, and in the result part…. In Materials and methods, the authors used “we” as the subject a lot. In my opinion, change the subject to your test object.
And, the language must be approved.
Unfortunately, my professional opinion about is work is not suitable to be published for now, and reject.
Reviewer 3 Report
The article “Hurdle effect of brine salting and high pressure processing on microbial inactivation and quality improvement in mackerel fillets” is well designed. This article can be reconsidered after revising the following comments:
Comment 1: Line 48 to 49: If the HPP can damage the tertiary structure, it obvious can affect the gelling properties of product.
Comment 2: Line 52 to 54: Besides these disadvantage? Why the authors recommend this HPP technology. As mentioned earlier it is good for nutrients but not for texture. In the meantime, both should be normal for consumer satisfaction.
Comment 3: Line 56 and 57: There are a lot of studies available on the NaCl, where is the novelty of this study.
Comment 4: The objectives and need of this study must be revised.
Comment 5: Line 96: why the authors choose this brine concentration?
Comment 6: Line 103: Time of carrying fish to Lab?
Comment 7: Line 111: why the authors choose these MPa treatments?
Comment 8: Section 3.1: what could be reason for decline in microbial count effectively at 300 and 400 MPa, need to add the appropriate discussion.
Comment 9: Protein aggregation could also reduce the textural properties, need to add this information.
Comment 10: Section 3.2: Conclude this section with better concentration of brine solution and HPP treatment.
Comment 11: Add the practical significance of this study, limitation and future work in the conclusion part.
Comment 12: What authors suggest to introduce this technique at industrial level to replace the conventional method.
Round 2
Reviewer 3 Report
All the comments are well addressed.